# Replication-associated inversions are the dominant form of bacterial chromosome structural variation

Matthew D'Iorio[1] , Ken Dewar[2,3]

The structural arrangements of bacterial chromosomes vary widely between closely related species and can result in significant phenotypic outcomes. The appearance of large-scale chromosomal inversions that are symmetric relative to markers for the origin of replication (*OriC*) has been previously observed; however, the overall prevalence of replication-associated structural rearrangements (RASRs) in bacteria and their causal mechanisms are currently unknown. Here, we systematically identify the locations of RASRs in species with multiple complete-sequenced genomes and investigate potential mediating biological mechanisms. We found that 247 of 313 species contained sequences with at least one large (>50 Kb) inversion in their sequence comparisons, and the aggregated inversion distances away from symmetry were normally distributed with a mean of zero. Many inversions that were offset from dnaA were found to be centered on a different marker for the *OriC*. Instances of flanking repeats provide evidence that breaks formed during the replication process could be repaired to opposing positions. We also found a strong relationship between the later stages of replication and the range in distance variation from symmetry.

## Introduction

Bacterial chromosomes are continuously reorganized by an interplay between mutations, lateral transfer, and mobile genetic elements. Mutations are often broadly categorized as either local, a point mutation affecting a single nucleotide, or global, a structural variant (SV) or rearrangement of an entire locus in the form of an inversion, duplication, insertion, deletion, or translocation. Local mutations are more readily identifiable because they can be detected by comparing relatively short DNA sequence segments, whereas the discovery of larger SVs often requires longer draft genomes or complete genome sequences to identify and categorize the full extent of the polymorphism (Periwal & Scaria, 2014).

Completed genome assemblies have become fundamental research tools throughout the life sciences. Spurred by multiple generations of massively parallel DNA sequencing technologies, complete genome sequences provide the framework for mutation discovery and can elucidate the mechanisms that influence bacterial evolution. Over the last decade, the continually increasing access to complete genomes offers an unparalleled opportunity to study structural variation across phylogenies and further elucidate its significance in modulating genotype and phenotype variation. Identifying and documenting SVs has been recognized as a promising method for tracking bacterial evolution and providing evidence for the underlying mechanisms that drive recombination (Noureen et al, 2019; Weigand et al, 2019).

Chromosomal rearrangements often result from homologous recombination, which is the process that contributes to the repair of DNA damage and resolves breakdowns that occur during replication (Kuzminov, 1999; Michel et al, 2007). Circular bacterial chromosomes generally replicate using two replication forks that move bidirectionally away from the replication origin (*OriC*) and conclude by meeting at the terminus (*Ter*). The position of these macrodomains is dynamic during replication, and their position and compaction are regulated by nucleoid-associated proteins and topoisomerases (Dillon & Dorman, 2010). Either or both replication forks can break down during regular growth conditions, and their resolution is essential for survival (Cox et al, 2000). A breakdown in both replisomes that is resolved by recombination at a single complex would result in an inversion (Makino & Suzuki, 2001), and if both replisomes are moving at the same speed, the inversion will appear symmetric relative to the origin of replication (Fig 1). These symmetric, "X"-shaped inversions are henceforth referred to as replication-associated structural rearrangements (RASRs).

RASRs are highly prevalent forms of SV in well-studied bacteria and were first identified by studying the conservation of the order of genes and the effects of disruptions to the conserved gene order (Sanderson & Hall, 1970). The significance of RASRs has been established because of their ability to alter phenotypic outcomes such as virulence, survivability, and pathogenicity of eukaryotic hosts (Hughes, 2000; Merrikh & Merrikh, 2018). For instance, a RASR was observed in isolated colonies of *Staphylococcus aureus*, which

---

[1]Quantitative Life Sciences, McGill University, Montreal, Canada    [2]Department of Human Genetics, McGill University, Montreal, Canada    [3]Centre for Microbiome Research, McGill University, Montreal, Canada

Correspondence: matthew.diorio@mail.mcgill.ca

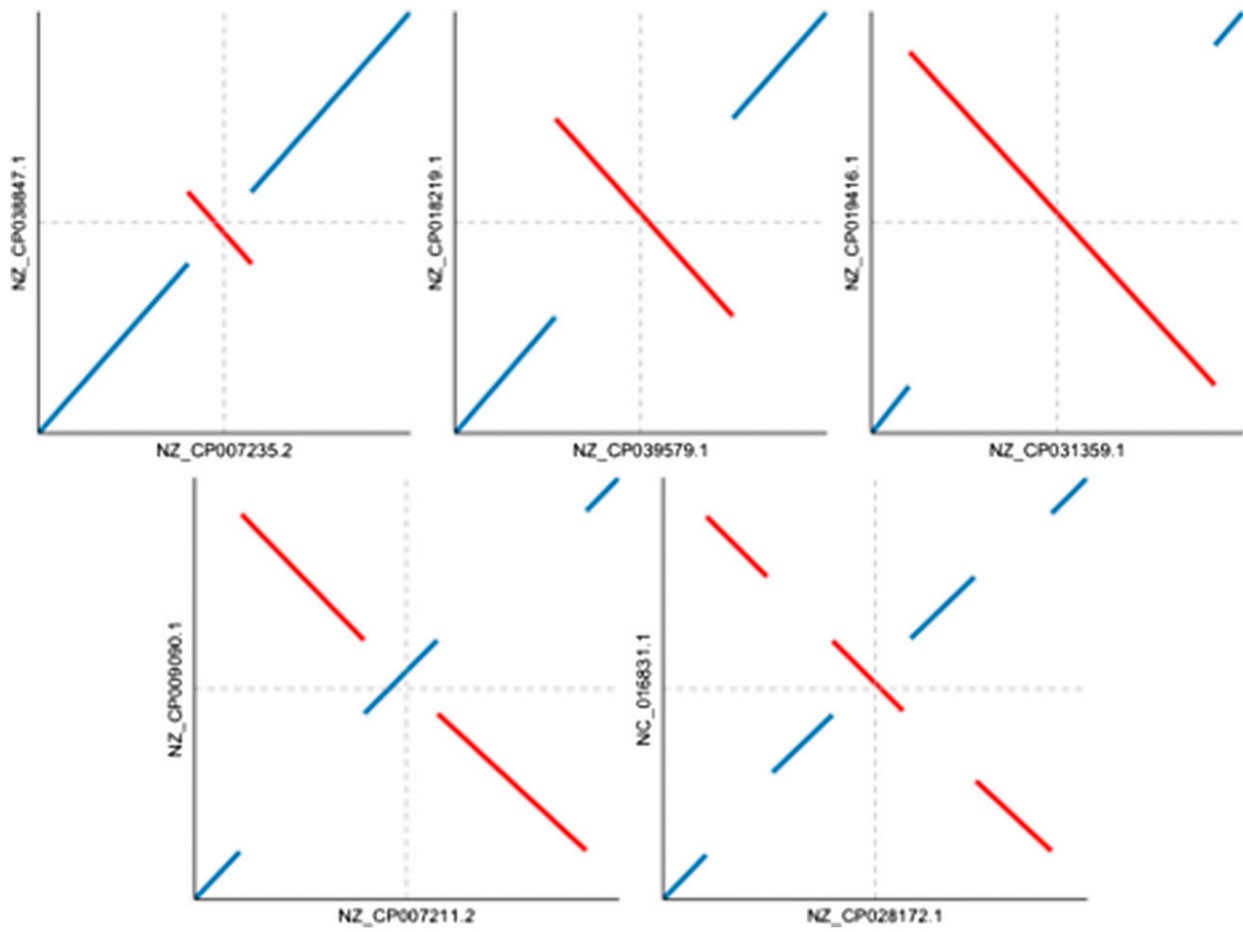

**Figure 1. Pairwise comparisons of *Salmonella enterica* sequences showing examples of replication-associated structural rearrangements (RASRs).**
Each reference is plotted on the x-axis, and the query is plotted on the y-axis. The horizontal and vertical arrows indicate the approximate position of the dnaA gene for the query and reference sequence, respectively. The top panel illustrates single RASRs of different lengths, and the bottom panel shows examples of double and triple RASRs.

regulated the expression of multiple genes and caused phenotypic changes such as antibiotic and immune resistance, growth rate, and colony morphology (Cui et al, 2012). Another RASR spanning 1 Mb in a pathogenic line of *Streptococcus pyogenes* was identified at a 40% higher frequency in patient screenings after 5 yr and correlated with resurgence in severe infections in Japan (Nakagawa, 2003). Instances of RASRs have also been extensively documented in model pathogens such as *Salmonella enterica, Escherichia coli,* and *Yersinia pestis* (Roth et al, 1996; Darling et al, 2008). Although synteny is disrupted by these mutational events, the distance of each gene to the origin of replication remains well conserved in most observations (Eisen et al, 2000; Tillier & Collins, 2000; Repar & Warnecke, 2017). The varying length and location of chromosomal inversions that occur during replication may be caused by the timing of the recombination event and the position of the macrodomains. Two broad topological structures of chromosomes have been observed in bacterial chromosomes which could meaningfully impact the way recombination takes place during replication (Wang et al, 2014). For instance, the proportion of large, interreplichore inversions was found to be associated with the observed chromosome topology where replichores are mostly in close proximity during replication (Khedkar & Seshasayee, 2016). Identification of RASRs within and across bacterial species could help to discern chromosomal topologies and the positions of macrodomains at the time of recombination.

In this study, we have used the National Center for Biotechnology Information (NCBI) Microbial Genomes Database (O'Leary et al, 2016) to assess the amount of large structural variation present throughout bacterial phylogeny. Although we restricted our analyses to complete genomes (31,196), which represent only ~3% of the total number of sequenced bacterial genomes (1,246,316), it still allowed a survey of 313 species collectively representing multiple major bacterial clades. Notwithstanding that the databases are heavily skewed toward human and animal pathogens, we have still been able to survey across a wide phylogenetic distribution and a wide variety of ecological niches. Our results confirm previous species-level observations that symmetric inversions represent the predominant form of structural variation. The investigation of breakpoint sequences shows that these inversions are likely mediated by homologous recombination because of repetitive elements, and the stage of replication influences the degree of symmetry. These RASRs can occur iteratively in successive bacterial lineages, offering an

epidemiological tool for tracking strain evolution and distribution. Although rearrangement-based phylogenetic network mapping has been examined in a limited capacity, our results identify a broad range of species with sufficient data to create rearrangement networks.

# Results

## RASRs are identified across phylogenies

We assessed a total of 313 species that were represented by 10 or more complete genome sequences in the NCBI database and identified distinct structural rearrangements in pairwise comparisons of samples within species groups. When two genomes of the same species shared high sequence similarity and did not contain any large structural rearrangements, they were clustered into colinear groups, which we refer to as genoforms. A genoform in this work refers to a group of collinear chromosomal sequences within a species with at least 80% sequence similarity and without an inversion spanning more than 50 Kb. This size of inversion represents ~1% of the genome length and is clearly detectable in a pairwise comparison segment plot because genome lengths range within our dataset from 0.4 to 9.8 Mb with an average of 4.7 Mb.

We were able to identify multiple distinct genoforms in 247 of 313 species included in our dataset across nine major clades. The number of RASRs and distinct collinear clusters identified generally increased with the amount of sequence data available for each species, and some species appeared to have a higher propensity for incurring structural variation than others. Some species groups contained many fully sequenced genomes and little evidence of structural variation such as *Mycoplasma pneumoniae* (79 genomes, 1 genoform) and *Chlamydia trachomatis* (91 genomes, 1 genoform). Genome stability for both species could be related to their slower reproduction rates relative to other bacteria (Vieira-Silva & Rocha, 2010). In contrast, some species appeared to have a higher propensity for rearrangements such as *Y. pestis* (58 genomes, 49 genoforms), *Glaesserella parasuis* (23 genomes, 19 genoforms), and *Aeromonas veronii* (27 genomes, 22 genoforms); all of which have relatively fast reproduction rates and higher repetitive genome content compared with other species.

Although our data was dominated by mammalian pathogens, we were also able to identify RASRs in bacteria that occupy a range of biological niches. There were instances of RASRs in different probiotic species and non-pathogens such as in three of four *Bifidobacterium* species, all six *Lactobacillus* species, and *Streptococcus thermophilus* and *Streptococcus salivarius*. We also found symmetric inversions in bacteria from environmental samples and plant symbionts such as *Bradyrhizobium diazoefficiens*, *Phaeobacter inhibens*, and *Rhodopseudomonas palustris*.

Incidences of RASRs were observed throughout the major clades of our dataset. We adjusted for the overrepresentation of Pseudomonadota (previously Proteobacteria, [Oren & Garrity, 2021]) by partitioning their species at the class level to characterize occurrence across the major clades. We observed that these types of inversions can be iterative, leading to increasingly complex genome shuffling in successive strain lineages (Fig 1), and examples of

RASRs across bacterial phylogeny are displayed in Fig 2. In this report, we now use nomenclature which the International Committee on Systematics of Prokaryotes recently implemented to revise prokaryotic phylum names (Whitman et al, 2018; Oren & Garrity, 2021) but also display the previous nomenclature in brackets for each clade that has changed.

## RASRs are the predominant form of chromosome structural rearrangement

The symmetry of each inversion was evaluated as the proportionate distance from the dnaA gene set to position zero (analogous to 12 o'clock), where a value of zero would indicate maximum symmetry and positive and negative values indicate proportionate distance away from dnaA to the left and right, respectively. The locus for the terminus is directly opposite of the origin for most bacterial species (analogous to 6 o'clock). Proportional distance of a midpoint of an inversion was calculated here as the range from −0.5 to 0.5 to account for both the absolute distance and the direction that an inversion is offset from symmetry relative to the dnaA locus. Distance values were measured for all inversions present in a pairwise comparison between representatives from different genoforms with the highest nucleotide similarity. In the 247 species with 2 or more genoform groups, there was an average of ~10 distinct genoforms per species and an average of ~3 inversions per genoform comparison. Our dataset containing all inversions was overrepresented by species that contained disproportionately high counts of inversions such as *E. coli* (88,917 inversions), *Bordetella pertussis* (16,630 inversions), *and Klebsiella pneumoniae* (15,955 inversions). These were the top three most represented genomes and together comprised ~52% of the total count of inversions throughout the dataset. Because the same inversions can appear when a single genoform is compared with multiple genoforms, the overall set of distance measurements for all inversions will be redundant. The set of distance values is taken as an average of all inversions within a comparison to decrease the redundancy in this count. The overall mean is recorded for each species by using the average distance value in each comparison.

The overall average distance to the dnaA gene was zero, and 61% of species were within a 10% overall mean proportionate distance to the annotated dnaA gene. To identify the tendency of inversions throughout our 247 species, we recorded the average inversion midpoint value for each species and plotted the distribution on a circular representation of the bacterial chromosome (Fig 3). The average midpoint of an inversion was calculated as the mean between the central point of each inversion on the coordinates for both the reference and the query genomes. We used proportionate midpoint values as a relative value for genome length to normalize for the wide range of genome sizes.

## Variation in RASRs across species groups

To understand the overall distribution of inversion symmetry, it was informative to normalize for genome length and calculate the distance as a metric to the left and right of the origin as well as the absolute distance from symmetry. The distribution of symmetry distances within species indicates which species conform to or deviate from the overall trend of symmetry. Using the counts of

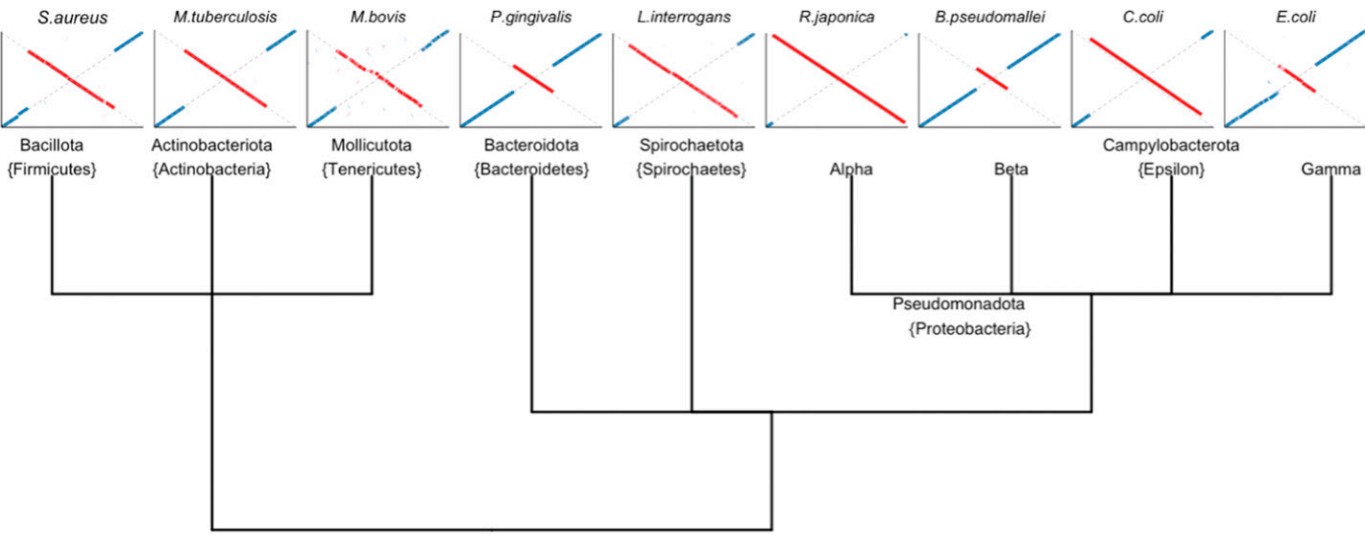

**Figure 2. Nine phylogenetic clades containing species with at least one large-scale structural variant.**
Replication-associated structural rearrangements were extensively documented throughout clades, and an example of a representative organism from each is displayed below each label. Structure of the phylogenetic tree is based on 16S similarity (Hug et al, 2016).

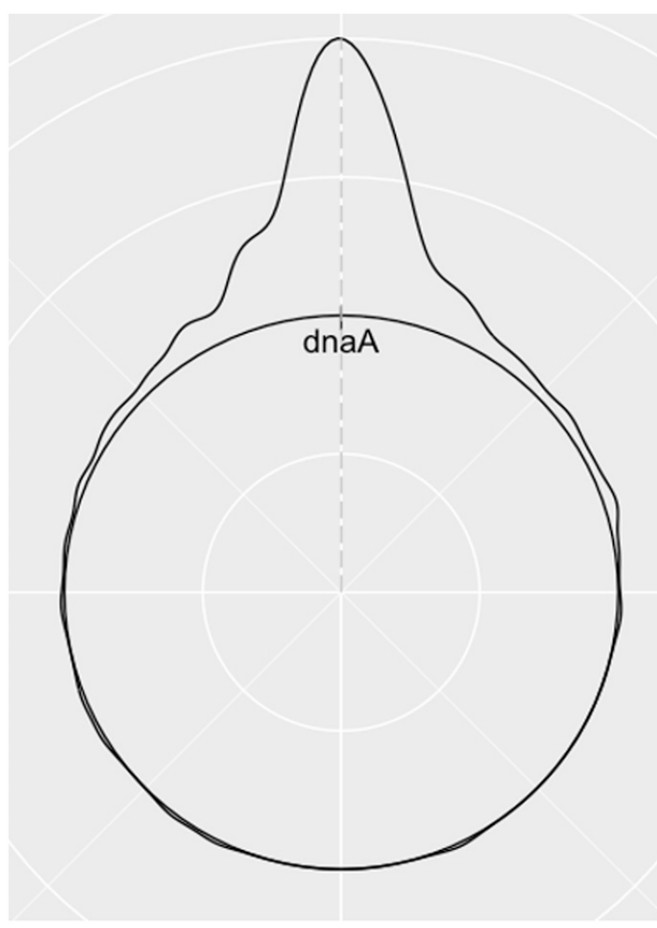

**Figure 3. Aggregate distribution of average midpoint values for all large-scale (>50 kb) inversions in each of the 247 bacterial species.**

inversions in species with at least 30 inversions and 5 distinct genoforms, we identified the distribution of symmetry distance in 92 species (Figs 4 and 5). Although the mode of distance for most species was at or very close to zero, we identified multiple species with bimodal distributions that were significantly offset from symmetry. In Gammaproteobacteria, the species *Haemophilus influenzae*, *Haemophilus parainfluenzae*, and *Mannheimia haemolytica* all contained inversions that were predominantly located away from the replication origin (Fig 4). We also observed species that were consistently offset from symmetry in one direction such as in *B. pertussis* (Fig 5). Because the dnaA gene is not always adjacent to the origin of replication, we identified the minimum distance between dnaA and the origin of replication annotated using the Z-curve method from the Doric database (Luo & Gao, 2019). Three possible dnaA to *OriC* distances were recorded for each species using the closest recorded Doric annotation to the dnaA loci: (1) negative values to represent *OriC* to the left of dnaA, (2) positive values to represent OriC to the right of dnaA, and (3) values of zero to represent *OriC* as directly beside dnaA. The distance from dnaA to *OriC* roughly corresponds to the distance we observed between dnaA and the mode of inversion midpoints.

## Homologous recombination could mediate RASRs during replication

Replication and recombination are interdependent processes of DNA metabolism (Syeda et al, 2014). Although replication forks move bidirectionally and independently along each side of the chromosome, the progression can be halted and the RecBCD pathway can initiate fork reversal to restart replication (Michel et al, 2004). Interspersed repeated segments may be mediators in the process of forming RASRs given that an arrest in replication could become resolved by reinitiating at opposite forks that share the same sequence. Repeated elements within the chromosome have

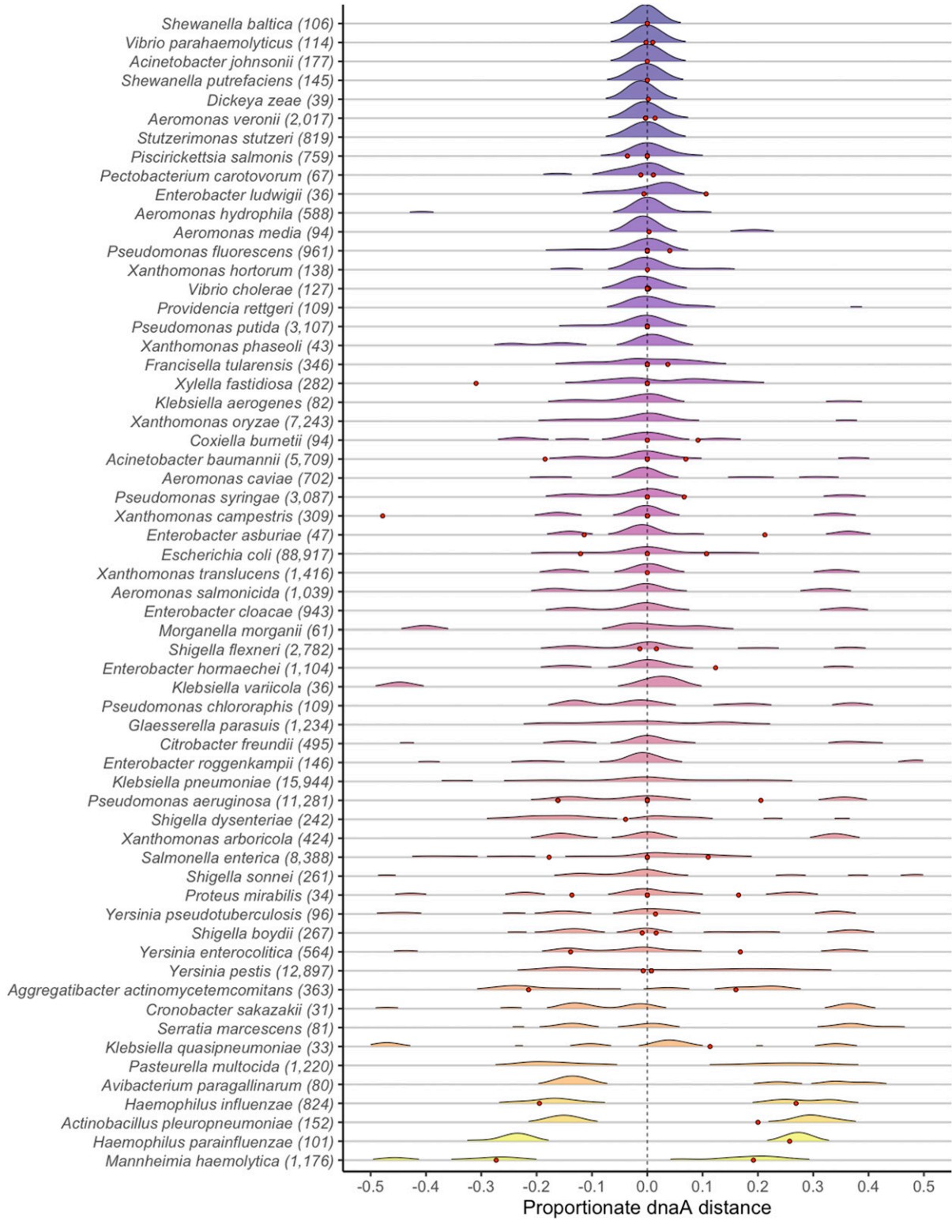

**Figure 4. Distribution of midpoints for 53 Pseudomonadota (Gammaproteobacteria) species.**
Point 0 represents the position of the annotated dnaA gene. The red points indicate the approximate positions of the closest origin available predicted by the Doric database (Luo & Gao, 2019).

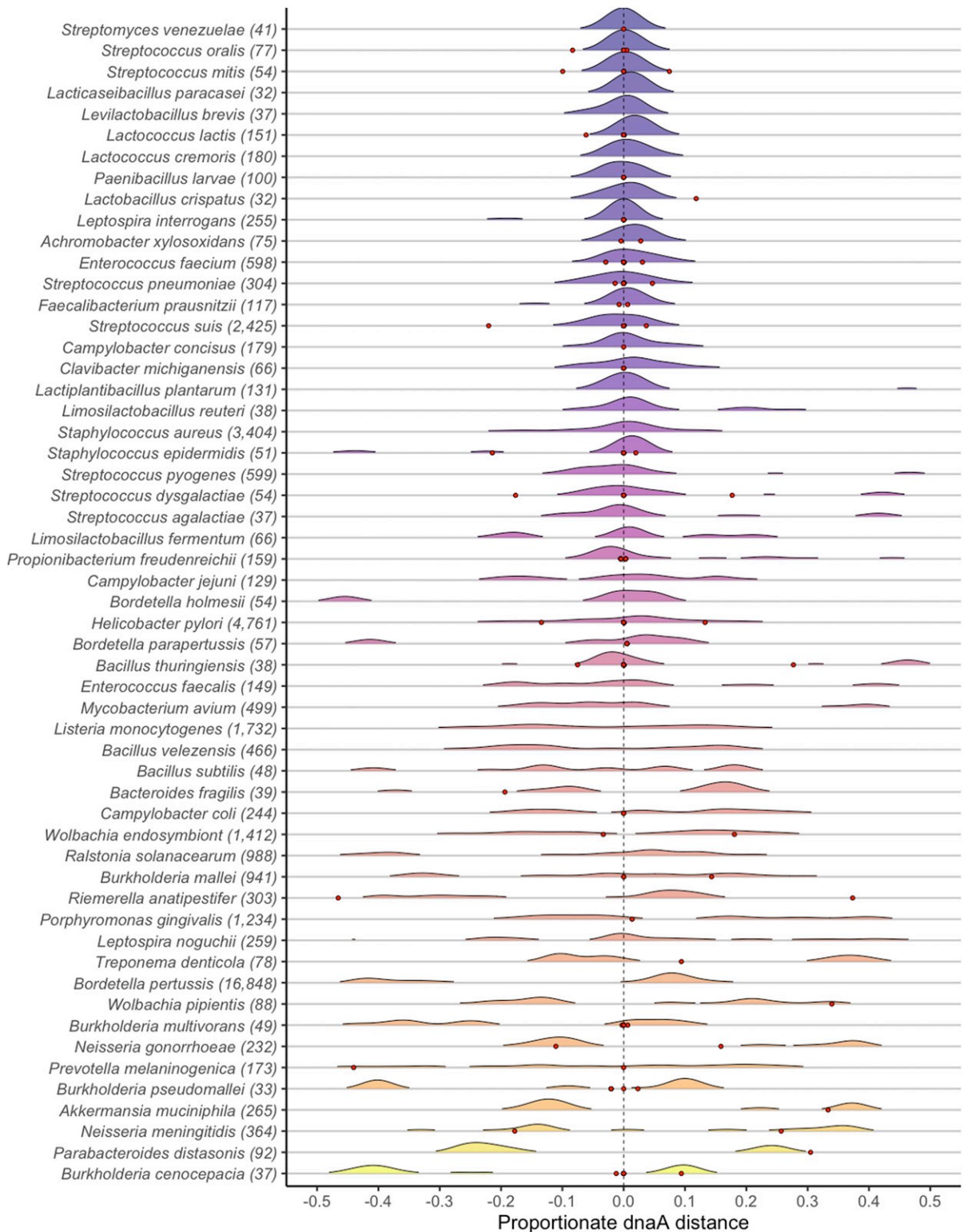

**Figure 5. Distribution of midpoints for 48 species from all clades except for Pseudomonadota.**
Point 0 represents the position of the annotated dnaA gene. The red points indicate the approximate positions of the closest origin available predicted by the Doric database (Luo & Gao, 2019).

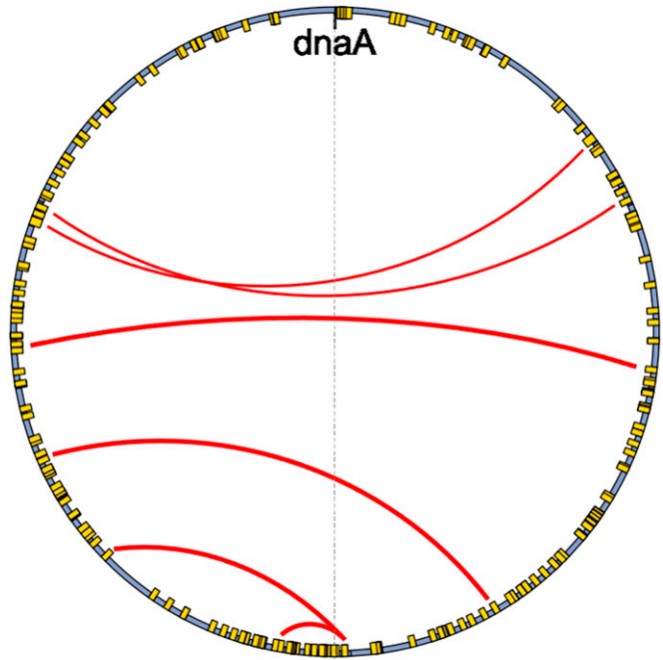

**Figure 6.   Inversion breakpoint locations (red lines) and repeated elements (yellow) found at breakpoints identified in the *Bordetella holmesii* reference chromosome (blue).**

been found to mediate chromosomal structural recombination (Achaz et al, 2003).

To investigate the potential relationship between repeat content and inversion occurrence, we calculated the mean proportion of repeat content for each species and compared that to a count of genoforms per the number of species. We found a low overall association ($r^2 = 0.14$) between repeat content and genoform proportion using linear regression. The overall proportion of repetitive content, and the total count of identified transposases within a sequence, did not correlate strongly with a propensity for a greater count of inversions or distinct genoforms relative to the number of sequences within a species group.

The role of repeat-induced mediation of inversions was further examined by identifying sequences involved in well-defined breakpoints observed in comparisons made between highly similar sequences from different genoform clusters. We found that the large inversions present within species groups were flanked by the same repetitive sequence on both ends, and this sequence was often highly abundant throughout the genome. For instance, there were six individual inversions with high-identity breakpoints found between clusters in *Bordetella holmesii*, and the sequence spanning the overlapping breakpoints was the same on both sides (Fig 6). Breakpoint sequences were abundantly identified throughout the *B. holmesii* reference genome and were identified as containing genes that code for transposases. The symmetry and repetitive breakpoints in the inversions found within *B. holmesii* were also previously noted and identified by Weigand et al (2019).

### Desynchronization causing asymmetry in inversion formation

The independence of replication forks is exemplified by the observations that unequal replisome distance from the origin is

commonly observed throughout the replication process and that one replication fork may be halted, whereas the other continues (Breier et al, 2005). Fluorescent labeling of live *E. coli* cells has provided evidence that sister replichores operate independently and the distance between them varies throughout the process of replication (Reyes-Lamothe et al, 2008). If large-scale chromosomal inversions occur within progenitor strands during replication, then an inversion that is offset from symmetry could be caused by an unequal speed of synthesis between replication forks. The offset distance relative to the dnaA locus caused by unequal replication time would be exacerbated toward the end of replication. The breakpoints of a RASR relative to the dnaA gene can be used to infer the relative location of replichores and, by extension, the amount of sequence that has been replicated at the time an inversion occurred. To identify the distribution of inversion midpoint distances relative to their corresponding breakpoint positions, we isolated the 44,186 pairwise comparisons that contain only a single large inversion. The proportionate distance from the dnaA locus to each breakpoint was measured for each comparison by subtracting the proportionate inversion length from one; this value represents the amount of replicated sequence to provide an index showing how far replichores progressed. These values were plotted against their corresponding dnaA distances (Fig 7A).

We identified an even greater tendency toward symmetry in this dataset given that 76% of individual inversions were within a 0.1 proportional distance of dnaA. We also found that the overall range of distances increased as replication progresses with roughly equal counts of inversions on either side of dnaA; 47% were negative (offset to the left), and 53% were positive (offset to the right). Distance ranges were calculated by grouping data by a sliding window along the replication index by gathering the values at every 0.01 increment. Inversions were observed at a similar frequency until roughly 90% of the way into the replication index, at which point the frequency sharply increased (Fig 7B). The linear model was then generated comparing the maximum and minimum value groups with the average replication index for each group of observations, where maximum values correspond to distance from dnaA to the right, and minimum values correspond to distance to the left. The maximum absolute range of values increased at a similar rate on each side of the sequence. The average time against maximum distance corresponded with further distance over the proportionate replication index at a rate of 2.04 ($r^2 = 0.83$, $P \ll 0.01$), whereas the average time against the minimum distance decreased at a rate of 1.66 ($r^2 = 0.76$, $P \ll 0.01$).

## Discussion

In this study, we have investigated the prevalence and potential mechanisms of chromosomal inversions symmetric to the origin of replication using NCBI's full collection of complete genome sequences. The overall prevalence of RASRs is evident from the high frequency of inversions that are observed to be centered near or directly on the origin of replication. The predisposition for structural variation can be influenced by the proportion of repeat content, given the evidence that an interspersed repeat was often

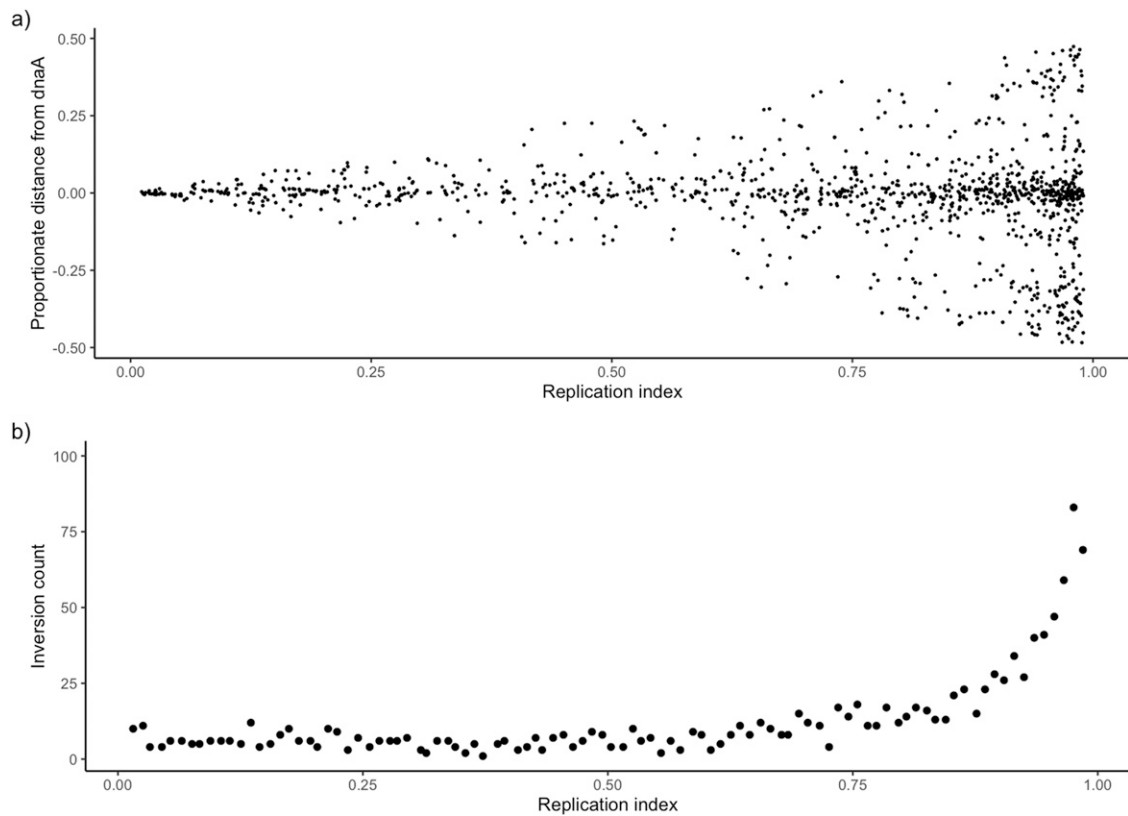

**Figure 7. Calculation of 44,186 dnaA distance values and inversion lengths from pairwise comparisons with only a single inversion with a length of at least 50 Kb.** **(A)** Inversion distance values calculated for all possible comparisons with an individual inversion plotted against the inferred replication time by using the proportionate inversion length **(A)**. For instance, a pairwise comparison with a small portion of the sequence inverted including the terminus would appear near the end of the replication index. **(B)** Count of inversions for a sliding window of 0.01 along the replication index shows the frequency of inversions as replication progresses (B).

found at inversion breakpoints. A strong tendency for inversions to congregate around the origin of replication is consistent with previous models that have suggested that inversions can be the result of breaks that form during the replication process. Although the overall prevalence of repeat content was not always associated with a higher incidence of inverted segments, equally interspaced repeated sequences appear to be an important component of the complete mechanism driving structural variation. Iterative inversions have been previously used to identify phylogenetic networks between sequences within species groups that have been limited to more specific species groups (Weigand et al, 2019), and the provided protocols and datasets in this study could be extended to identify phylogenetic networks across all bacterial species that have multiple complete sequences available.

We also note multiple lines of evidence excluding the possibility that the appearance of large-scale symmetrical inversions is simply a result of technical errors in DNA sequencing or computational assembly. First, given the wide range of species, sequencing centers, sequencing dates, and sequencing strategies that have been used, this consistent pattern of genetic diversity is highly unlikely to be the result of technological errors. Second, the observation of having these inversions centered on dnaA indicates coordination of breakpoints, again making a sequencing error unlikely. Third, in instances where we can recover genome assemblies and the long

reads underlying them, we can conclusively show that multiple independent reads clearly demonstrate that the inversion is a biological difference and not an assembly artifact. Conservation of inversion sizes and evidence of iterations over generations suggest that the propensity for symmetric chromosomal inversions is a basic form of genetic variation common to all types of bacteria.

The strong overall tendency for inversions to be symmetric to the origin of replication provides evidence that the underlying mechanisms are driven by breakage and misrepair during the replication process. DNA supercoiling before and after replication forks may be the cause in the formation of a single complex wherein crossing over of replichores along with double-ended breakage and repair ultimately forms the inversion. If both break sites contain identical sequences (i.e., interspersed repeats), then there should be an increased likelihood of misrepair to opposite ends. A single daughter chromosome also has both a leading and lagging strand synthesizing at each replichore, and our observed shifts away from symmetry may be a direct result of the difference in speed of synthesis. Moreover, because the termination site of replication is often but not always 180 degrees away from the origin, the instances where the termination site is asymmetrical could influence the axis of an inversion (Kono et al, 2014). This range of possible errors is exacerbated if the break occurs at a later stage in the replication process. The even range of distance from either side of

the replication origin as an overall trend suggests that the error rate in both replication forks move at a similar speed with a comparable error rate.

The pervasiveness and iterative nature of these RASRs underscore the potential for this method of clustering to be used as a method for tracking bacterial evolution. Structural variation analysis could add a useful dimension for phylogenetic analysis that could complement current methods such as nucleotide and gene content similarity. Mapping networks of genome rearrangements have been created in for specific groups of bacteria such as the *Bordetella* genus (Weigand et al, 2019); however, the increasing amount of complete genome sequence data will further the ability to conduct this type of analysis across more species and with greater depth. Many of the organisms analyzed here represent species groups with sufficient complete sequence data and multiple structural rearrangements. The table provided in our GitHub page can also serve as a resource for identifying which species groups could be used to generate phylogenetic networks as part of a broader analysis. Full-length genomes still represent a very small proportion of publicly available bacterial sequence data, given the resources and costs associated; however, their potential to aid in the discovery of the intricacies of bacterial diversity has not yet been fully realized.

# Materials and Methods

### Data collection

Sequence data were downloaded directly from the NCBI RefSeq and GenBank public datasets on (02 August, 2022). All RefSeq and GenBank assembly summaries were combined and filtered to generate a unique table containing all RefSeq and nonredundant GenBank summaries. Species with fewer than 10 available sequences were omitted. The corresponding FASTA and GFF files were downloaded for all other assemblies that included an annotated dnaA gene. We also used data from the Doric 10.0 bacterial database (Luo & Gao, 2019), which was last updated (17 June, 2018). The scripts that were created to download and process this data for analysis and generation of figures are available at https://github.com/mdiorio371/RASR. The GitHub page also includes a supplementary table showing each species analyzed, as well as the number of sequences, genoforms, and inversions within each species group.

### Synchronization to dnaA

NCBI submissions of circular bacterial genomes may not have standardized starting points and orientations. Because the dnaA gene is ubiquitous in bacteria and is a marker for the position of the origin of replication, we used it as a starting point for pairwise comparisons of a linearized representation of the genome sequences. We synchronized every genome sequence by using the coordinates of the dnaA gene from each assembly GFF file to reorient and rotate each sequence as necessary to create the same starting position and polarity for every

sequence. We developed a standalone webtool to synchronize individual genomes that is freely and publicly available at https://www.computationalgenomics.ca/tools/bacterial-genome-synchronizer/. Genomes that did not have an annotated dnaA sequence, and those with multiple dnaA sequences, such as those within Chlamydiae, are currently omitted from the pipeline.

### Pairwise alignment and collinearity clustering

After synchronization of each genome, all sequences were pairwise-compared with all other members of the set of sequences corresponding to the same species. We used MASH (Ondov et al, 2016) to provide a nucleotide similarity score and penalized the score when genomes were different lengths by subtracting a normalized value of the sequence length difference. We used this measurement to select the genome that had the highest similarity to all other genomes within the species as the starting point used to cluster for collinearity. This sequence was then used as the reference sequence that was aligned to all other sequences within the species using MUMmer4 (Marçais et al, 2018). Any sequence that did not show an inversion at a minimum length of 50 kb and had at least 80% of the reference covered by alignments was considered collinear. The minimum inversion length 50 Kb was useful for several reasons: (1) it is longer than most individual read lengths of long-sequence technologies, which is roughly 30 Kb for technologies such as single molecule, real-time, and nanopore sequencing (Amarasinghe et al, 2020), and (2) it will exclude most bacteriophages because the their most common genome size is between 40 and 45 Kb (Zrelovs et al, 2020). Inversion length of 50 Kb is also visually identifiable for all genome lengths in a standard pairwise comparison plot given that most complete sequences in our dataset were roughly 5 Mb. This step is repeated for all noncollinear sequences until each was sorted into a collinear cluster. After sequences were sorted, each cluster was pairwise-compared using the two representative sequences that had the highest shared nucleotide identity. Groups of collinear sequences were alphabetically named and referred to as "genoforms," to signify that the sequences in that group shared a similar genomic form.

Coordinates for an inversion's center, breakpoints, and distance to the dnaA locus were found by taking the average of coordinates in the reference and query and then dividing by the average of the two genome lengths to provide a proportional value. Proportional distance from the dnaA locus was adjusted from the range of (0, 1) to (−0.5, 0.5) to account for the circularity of the genome. A value close to zero represents inversion symmetry, and the absolute value represents the distance that the inversion offset from dnaA. The average distance for all inversions within each comparison was calculated to achieve an overall distribution of distances across species that would also decrease the redundancy of counting inverted loci that were present in multiple comparisons. The complete list of all inversion distances was also used to identify where offset inversions were distributed within species groups with a larger sample of inversions.

We also determined the proportion of each sequence that was repetitive to measure the association between repetitive content on genome stability. This was performed by counting the total number of repeated k-mers at a length of 31 nucleotides and

dividing by the total number of possible 31-mers based on the full length of the genome. We used the *kmercountexact.sh* function from bbmap to count the number of duplicated 31-mers (Bushnell, 2014). The 31-nucleotide threshold was chosen to capture the maximum amount of sequence at the maximum speed chosen given the recommendation within the bbmap documentation. Speed was an important factor because the mean repetitive value was calculated for all genome sequences that were part of a species group with at least two distinct colinear groups (42,631 sequences). This measurement correlated well with a prokaryotic database of repetitive measurements using a global calculation of sequence repetitiveness (Haubold & Wiehe, 2006).

The prevalence of inversions was quantified by the number of genoforms divided by the number of sequences within a species group. If multiple sequences were submitted from the same sequencing center, technology, and year, and were in the same genoform, then only one of the sequences was counted. This was performed to reduce the amount of redundancy in the total sequence count for species that were often nearly identical.

## Data Availability

All code and data can be found at https://github.com/mdiorio371/RASR.

## Supplementary Information

## Acknowledgements

We are indebted to the NCBI microbial genomics databases. We are deeply grateful to the multitudes of sequencing centers and personnel for their efforts in generating high-end genome sequences and their commitment to free and open access data sharing. We thank Romain Grégoire and the Canadian Centre for Computational Genomics for help in designing and hosting our genome synchronization application. We thank professors Greg Marczynski, Mathieu Blanchette, and Rodrigo Reyes-Lamothe for advice, guidance, and critical reviews of earlier versions of this work. M D'Iorio is supported by the McGill Genome Centre and Genome Canada.

### Author Contributions

M D'Iorio: conceptualization, data curation, software, formal analysis, investigation, visualization, methodology, and writing—original draft.
K Dewar: conceptualization, resources, formal analysis, supervision, funding acquisition, investigation, methodology, project administration, and writing—review and editing.

### Conflict of Interest Statement

The authors declare that they have no conflict of interest.

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
