## [Reviewer comments · Life Science Alliance]

Life Science Alliance

Replication-associated inversions are the dominant form of bacterial chromosome structural variation

Matthew D'Iorio and Ken Dewar

DOI: <https://doi.org/10.26508/lsa.202201434>

Corresponding author(s): Matthew D'Iorio, McGill University

Review Timeline:

Submission Date:	2022-03-04
Editorial Decision:	2022-04-08
Revision Received:	2022-09-08
Editorial Decision:	2022-09-27
Revision Received:	2022-09-29
Accepted:	2022-09-30

Scientific Editor: Novella Guidi

Transaction Report:

April 8, 2022

Re: Life Science Alliance manuscript #LSA-2022-01434-T

Matthew D'Iorio
McGill University

Dear Dr. D'Iorio,

Thank you for submitting your manuscript entitled "Replication-associated inversions are the dominant form of bacterial chromosome structural variation" to Life Science Alliance. The manuscript was assessed by expert reviewers, whose comments are appended to this letter. We invite you to submit a revised manuscript addressing all the Reviewer comments.

Thank you for this interesting contribution to Life Science Alliance. We are looking forward to receiving your revised manuscript.

Sincerely,

B. MANUSCRIPT ORGANIZATION AND FORMATTING:

Reviewer #1 (Comments to the Authors (Required)):

In this manuscript, the authors systematically identified replication-associated structural rearrangements of complete-bacterial genomes in NCBI Microbial Genomes Database, and they found most of large inversions are related to the locations of *dnaA* gene and replication origins, suggesting replication fork arrest may be a mechanistic cause for the asymmetry in some inversions. The results in this study are detailed and reliable, and the conclusions of the manuscript is convincing. I enjoy this manuscript, which is interesting, well written and easy to read. However, I tried to visit their standalone webtool to synchronize individual genomes several times provided in this manuscript, and it is always inaccessible. I suggest the authors improve the network and server capabilities. It will be better if the authors could make their data and pipeline available at GitHub or other public platform.

Reviewer #2 (Comments to the Authors (Required)):

Manuscript "Replication-associated inversions are the dominant form of bacterial chromosome structural variations" provides a nice description of patterns of large-scale inversions across bacterial taxons. Note that over-representation of symmetrical inversions, which basically constitutes the manuscript title, was previously shown (in the classic works Eisen 2000 and Darling 2008). The paper provides a complete overview of inversion patterns in bacterial world using all the available bacterial species data in GenBank and discussed the ecological factors affecting them.

Major revisions

0. No technical details behind the figures are provided. I firmly believe that the research has to be open and reproducible. Therefore, I would strongly suggest to make both the generated datasets and the computational pipelines publicly available. At the very least, the data underlying Figures 3 and 4 must be available somewhere online for other researchers.

1. The thresholds used in the research require motivations. In particular:

- an inversion at a minimum length of 50 kb;
- k-mers at a length of 31 nucleotides.

While the specific values may or may not be reasonable, it is unclear from the manuscript, why these were taken, and how robust the results are to these values.

2. Figure 5a. I think the idea of illustrating the relationship between the repeat content and large structural rearrangements is great. Indeed, I would expect to see the correlation as rates of inversions are higher in species with high number of insertion sequences in genomes (see Seferbekova et al. 2021, Bochkareva et al. 2018). However, I am afraid that the figure as it is does not provide enough statistical evidence. It is not quite clear what "significant" means (I would expect to see a specific p-value), and the resulting R squared of 0.13 seems pretty low to me. (Intuitively, the cloud of points does not obviously clusters around some upward-sloped line in the figure.) Also, it would be interesting to make sure this result is robust with respect to the k-mers threshold mentioned above.

3. Section 'De-synchronization causing asymmetry in inversion formation' would greatly benefit from a major rewrite, to make it easier to understand.

In Figure 6 it is not obvious what are exact values along the axes. Do I understand correctly that one point corresponds to a single inversion? If so, how was the replication index calculated? Is it just the inversion length? Maybe the illustration of calculated values on the circular chromosome would make the graph more readable?

4. I would suggest two revisions to the Discussion section.

First, I would expect some comparison of the results with other papers focusing on specific species.

Second, the authors note that "structural variation analysis could add a useful dimension for phylogenetic analysis." Indeed, there is a set of algorithms for phylogenetic reconstruction based on different types of rearrangements (Alekseyev MA, Pevzner PA. 2009; Tang et al 2014) and there are papers where the efficiency of this approach were discussed (Miklos and Darling 2009,

Bochkareva et al, 2017). However, since there is already a vast research incorporating the structural variation into phylogenetic analysis, I would suggest to either eliminate the note, or to specify what exactly the authors propose in this regard.

Minor revisions

1. text formatting here (Clark et al. 2016 (O'Leary, 2016 #66))
2. italic for bacterial species in figures
3. Legend of Fig 3. Please, add how many inversion are shown.

Reviewer #3 (Comments to the Authors (Required)):

In this manuscript, D'Iorio and Dewar consider inversion patterns in bacterial genomes. Early (Eisen et al 2000; Tillier and Collins 2000) and more recent work (Repar & Warnecke 2017; Khedkar & Seshasayee 2016) has established that there is a tendency, often strong, towards symmetric inversions in a variety of bacterial clades. Here, the authors assess inversions between bacterial genomes that share the same species designation. Because these genomes are closely related, they are more likely to align across a sizeable portion of the genome, facilitating the inference of inversions, including nested inversions. Khedkar & Seshasayee (2016) had similarly focused on closely related genomes (based on rRNA similarity rather than species labels) and the results reported here are overall very similar to those found in Repar & Warnecke (2017) and Khedkar & Seshasayee (2016) in that they show that, across diverse bacterial clades, inversions tend to be symmetric relative to the origin-terminus axis (here approximated via the location of *dnaA*). While I have not learned much new, the analytical approach is sufficiently distinct and focuses on a different set of species, that it can be seen as valuable independent corroboration of these prior studies.

Major comments:

- while some of the earlier work is mentioned in the introduction, key work is missing (in particular Khedkar & Seshasayee 2016, PMC4889656) and the authors could do a much better job of discussing their results in the context of prior knowledge.
- the manuscript lacks some methodological detail. In particular, I was left wondering how exactly breakpoints were determined (via some pre-existing software? custom script? manually?).
- a table with the species and the corresponding number of complete strains analyzed, inferred genoforms and inversions should be provided as part of the main text, with a list of strain IDs as a supplementary table.
- inversions appear to be counted via pairwise comparisons between genoforms. This is clearly the easiest way but comes with a drawback: not all these inversions will represent unique events. Imagine an inversion that happened early in the split of 5 strains (lineage A) from 5 others (lineage B). A second inversion then happened in lineage B, leading to 3 genoforms: A, B1, and B2. As I understand the authors' approach, the initial inversion would be counted twice, in the comparison of A-B1 as well as A-B2. If this is correct, then it is hard to understand what the number of inversions reported (127,161) actually means and how many evolutionarily unique events there are. Presumably it's fewer than ~6 per genome (from dividing 127,161 by the number of , genomes analyzed: 21,856 genomes)? Given the number of strains involved, 127,161 seems rather high. This way of counting also means that species with more strains and/or higher rates of inversion would be associated with a disproportionately higher number of events. The authors should clarify their counting approach, either determine/estimate the number of events or appropriately contextualize the numbers reported here.
- regarding the analysis of repeat elements and their role in facilitating homologous recombination, I have three comments:
 - o the authors should relate their results to prior findings, notably in Eisen et al (2000), who detected repeats equidistant from the origin when they used Mummer to compare a given genome with itself, and in Khedkar & Seshasayee (2016) who consider inter-replichore interactions in the context of 3D genome structure
 - o The abundance of interspersed repeats might indeed be taken as a predictor of the propensity to recombine, but in the context of this study, the location of those repeats also matters (equidistance from origin along both replichores); could a more focused test be done here?
 - o the overall evidence presented specifically in this section provides only weak support for homologous recombination during replication as a driving factor and does not warrant a subheading as strongly worded as "RASRs are mediated by homologous recombination during replication"
 - o I am not sure what conclusion I am meant to draw from Figure 6. What is the expectation here? Am I correct in assuming that the distribution of distances from *dnaA* will by necessity be heteroscedastic inversions that involve breakpoints further away from *dnaA* will be larger and therefore allow more variability in the distance to *dnaA*?

Minor comments:

- I appreciate that the cut-off for how to define a "large" inversion is necessarily arbitrary. I nonetheless wonder what motivated the cut-off at 50kb. Was there some empirical pattern that suggested 50kb as a reasonable cut-off?
- p.6 in the sentence "The tendency towards..." delete the "the" before "proportionate distance"
- in the following sentence "Pertussis" should be lowercase

- Figure 4: it would be helpful to provide the number of inversions along each species name to illustrate how many inversions those distributions are based on
- regarding symmetric/asymmetric progression of the replication forks, we provided some preliminary data on why some species might be more prone to symmetric breaks than others, namely those where forks tend to be co-localized during replication, suggesting communication between complexes so that one might respond when the other stalls (see Fig 5 in Repar & Warnecke 2017).
- the authors suggests that repeats on located opposite on either replicore might facilitate inversion. I think this is probably correct, but there is also the possibility (not mutually exclusive) that opposite repeats are the result of inversion events. A more formal investigation of repeat distributions along replichores with reference to oriC would be interesting in this regard.
- attempts to open <https://www.computationalgenomics.ca/tools/bacterial-genome-synchronizer/> ended in a blank browser window (tried with multiple browsers); please fix.

It is my policy to sign my reviews.

Tobias Warnecke

Letter to the reviewers

We would like to thank all the reviewers for their time and thoughtful comments that contributed to improving the quality of our manuscript. We have taken these comments into account and hope we have addressed each concern well in our revised manuscript. The pipeline we used has been revised and the code we used is now up on Github, we also reran the complete analysis using more up-to-date data from NCBI.

Reviewer 1

We are glad that you enjoyed reading the manuscript and hope that our revisions have improved the work further. Our standalone tool has been redeployed and we have added contact information to the page in case there are any future issues with the application. We have also uploaded the entire pipeline including all functions and data pre and post-processing to the GitHub page.

Reviewer 2

As mentioned in the previous response, the linked GitHub page contains the complete pipeline, and each figure can be reproduced using the script provided. Motivations for why the thresholds were chosen were added to the methods section. The 50 Kb threshold is a pragmatic choice because it is long enough to extend past most next-generation sequence read lengths and most phage lengths to limit smaller inversions that could be identified independently from the replication process. The 31-kmer length is recommended by the associated software's documentation to optimize speed since this process was applied to every complete sequence in our final dataset. To further verify our method for quantifying repetitiveness, we compared it to another index of repetitiveness dataset created by Haubold and Wiehe (2016) and found both correlated and achieved similar results when compared to the proportion of genofoms per species.

The section "Homologous recombination could mediate RASRs during replication" has been updated to clarify our results. We still discuss the slight association between repetitive content and the propensity for more RASRs that we found, but further emphasize that repetitive sequences are present in all inversions that we examined that had high-identity alignments flanking the breakpoints. The figure has been replaced to show the ~1kb or higher repetitive sequences that were found on each end of the inversions within one of our species groups. We also updated the section "De-synchronization causing asymmetry in inversion formation" with a more in-depth explanation of the results and methodology in the text and in the figure description.

The discussion and the introduction have been updated to include references to literature focusing on specific species, and how this type of analysis could be used to build phylogenetic networks.

Reviewer 3:

The introduction has been updated to include a discussion of the previous results from Khedkar & Seshasayee. The methods were also updated to include a description of how breakpoints were determined, and the function that was used to identify them is available as part of the associated GitHub page. Supplementary data including the number of complete sequences within a species group, inversions, and genoforms identified can be found in the data/curated folder as species_summary.csv. A better description of the method of counting inversions has been added, and more examples from the data have been provided to help contextualize the numbers reported in the methods and results. The average inversion midpoint is taken for each species to reduce the redundancy in counting iterative inversions for the section “In the section “RASRs are the predominant form of chromosome structural rearrangement“. Every inversion midpoint is identified in “Variation in RASRs across species groups” despite the redundancy to get a more detailed picture of how the distributions of all inversions look within each species group. The “De-synchronization causing asymmetry in inversion formation” section uses pairwise comparison data specified to only have a single inversion.

As mentioned in the response to reviewer 2, “Homologous recombination could mediate RASRs during replication” has a new figure and has been rewritten to add more context to emphasize the importance of the position of the repeat sequences rather than the overall repetitiveness of the genome.

More components were added to the figure to illustrate the idea and a greater explanation of the figure has been written in the description and the text. Justification for the 50 Kb threshold was added (more detail in response to reviewer 2). Adjustments have been made to each of the figures to add more information and clarity, and more context has been added to the introduction and discussion. As mentioned to reviewers 1 and 2, the complete pipeline is now available on GitHub and the webtool has been fixed. We also added contact information to the website in case it has future issues.

Thank you, Dr. Warnecke for your thoughtful comments and thorough reading of our paper.

September 27, 2022

RE: Life Science Alliance Manuscript #LSA-2022-01434-TR

Mr. Matthew D'Iorio
McGill University
740 Dr Penfield Avenue
Montreal, Quebec H3A 0G1
Canada

Dear Dr. D'Iorio,

Thank you for submitting your revised manuscript entitled "Replication-associated inversions are the dominant form of bacterial chromosome structural variation". We would be happy to publish your paper in Life Science Alliance pending final revisions necessary to meet our formatting guidelines. Please revise and format the manuscript and upload materials by Thursday.

- please consult our manuscript preparation guidelines <https://www.life-science-alliance.org/manuscript-prep> and make sure your manuscript sections are in the correct order and add a separate figure legend section
- please upload your main figures as single files; these will be displayed in-line in the HTML version of your paper, so please provide them as single page files (Figure 4 currently spans 2 pages); we do not have a limit on the number of main figures and these can be split if necessary for space
- on page 13, please double-check your figure callout for Figure 5b; Figure 5 doesn't have a panel B, and it seems like this should be a callout for Figure 6B

A. FINAL FILES:

B. MANUSCRIPT ORGANIZATION AND FORMATTING:

Thank you for your attention to these final processing requirements. Please revise and format the manuscript and upload materials by Thursday.

Sincerely,

Reviewer #3 (Comments to the Authors (Required)):

The revised manuscript contains useful clarifications of the methodology. Although I still wish the results, which are confirmatory in nature, had been interpreted with more explicit reference to prior work, this quibble should not be seen as a reason to hold up publication of the manuscript.

It is my policy to sign my reviews.

Tobias Warnecke

September 30, 2022

RE: Life Science Alliance Manuscript #LSA-2022-01434-TRR

Mr. Matthew D'Iorio
McGill University
740 Dr Penfield Avenue
Montreal, Quebec H3A 0G1
Canada

Dear Dr. D'Iorio,

Thank you for submitting your Research Article entitled "Replication-associated inversions are the dominant form of bacterial chromosome structural variation". It is a pleasure to let you know that your manuscript is now accepted for publication in Life Science Alliance. Congratulations on this interesting work.

DISTRIBUTION OF MATERIALS:

Again, congratulations on a very nice paper. I hope you found the review process to be constructive and are pleased with how the manuscript was handled editorially. We look forward to future exciting submissions from your lab.

Sincerely,
